# Identification of BRCC3 and BRCA1 as Regulators of TAZ Stability and Activity

**DOI:** 10.3390/cells12202431

**Published:** 2023-10-11

**Authors:** Silvia Sberna, Alejandro Lopez-Hernandez, Chiara Biancotto, Luca Motta, Adrian Andronache, Lisette G. G. C. Verhoef, Marieta Caganova, Stefano Campaner

**Affiliations:** Center for Genomic Science of IIT, CGS@SEMM (Istituto Italiano di Tecnologia at European School of Molecular Medicine), Fondazione Istituto Italiano di Tecnologia (IIT), 20139 Milan, Italy; silvia.sberna@iit.it (S.S.);

**Keywords:** YAP1, TAZ, WWTR1, BRCC3, BRCA1, post-translational regulation

## Abstract

TAZ (WWTR1) is a transcriptional co-activator regulated by Hippo signaling, mechano-transduction, and G-protein couple receptors. Once activated, TAZ and its paralogue, YAP1, regulate gene expression programs promoting cell proliferation, survival, and differentiation, thus controlling embryonic development, tissue regeneration, and aging. YAP and TAZ are also frequently activated in tumors, particularly in poorly differentiated and highly aggressive malignancies. Yet, mutations of YAP/TAZ or of their upstream regulators do not fully account for their activation in cancer, raising the possibility that other upstream regulatory pathways, still to be defined, are altered in tumors. In this work, we set out to identify novel regulators of TAZ by means of a siRNA-based screen. We identified 200 genes able to modulate the transcriptional activity of TAZ, with prominence for genes implicated in cell–cell contact, cytoskeletal tension, cell migration, WNT signaling, chromatin remodeling, and interleukins and NF–kappaB signaling. Among these genes we identified was BRCC3, a component of the BRCA1 complex that guards genome integrity and exerts tumor suppressive activity during cancer development. The loss of BRCC3 or BRCA1 leads to an increased level and activity of TAZ. Follow-up studies indicated that the cytoplasmic BRCA1 complex controls the ubiquitination and stability of TAZ. This may suggest that, in tumors, inactivating mutations of BRCA1 may unleash cell transformation by activating the TAZ oncogene.

## 1. Introduction

The yes-associated protein (YAP) and the WW domain-containing transcription regulator protein 1 (WWTR1/TAZ) are transcription factors historically identified as essential effectors of the Hippo-signaling pathway [1]. The Hippo pathway comprises a hierarchically organized core module comprising the MST1,2 and the LATS1,2 kinases, which, once activated, phosphorylate YAP/TAZ. The hyper-phosphorylation of YAP/TAZ results in their cytoplasmic retention and ubiquitin-dependent degradation by the SCF/ß-TRCP complex [2,3]. The upstream signals and pathways regulating YAP/TAZ are highly articulated and involve apical-basal polarity factors, cell-to-cell and cell adhesion proteins, glucose metabolism, G-protein coupled receptors, WNT signaling, and the heat-stress response [4,5,6,7,8,9,10,11]. These stimuli can regulate YAP/TAZ activity, both in a Hippo-dependent and Hippo-independent fashion.

Once activated, YAP/TAZ control gene transcription by binding the TEADs transcription factors, which bear sequence-specific binding to DNA but lack transactivating activity. The YAP/TAZ–TEAD dimers activate gene expression by binding both promoters and enhancers, thereby stimulating the epigenetic remodeling of target loci and favoring the recruitment of RNA polymerase, as well as its release from paused promoters [12,13,14,15]. The YAP/TAZ–TEAD complexes can also repress gene expression, although the precise mechanisms are less clear [12,16]. YAP/TAZ transcriptional activity are also boosted and shaped by the association to other transcription factors, which results in either potentiation of their canonical activity, as is the case for AP-1 [13], or extends their action on non-canonical targets as for MYC [17], HSF-1 [18], and other transcription factors [16]. Once activated, YAP/TAZ regulate the expression of genes linked to cell cycle, survival, cell growth, intracellular mechanical homeostasis, and dedifferentiation. This accounts for the reported activity of YAP/TAZ as promoters of cell proliferation during embryogenesis, tissue regeneration, and in cancer. Concerning cancer, YAP/TAZ activity has been linked to mesenchymal-like features, metastatic potential, chemo-resistance, and cancer stem cell properties [19,20]. In particular, TAZ is a prominent master regulator of such activities in basal-like breast cancers [21,22].

The BRCA1 gene encodes for an E3 ubiquitin–ligase belonging to the RING finger family. This protein is found to be associated with multiprotein complexes (BRCA–1 A–D complexes) that are involved in controlling genome stability by favoring repair for DNA double-strand breaks by homologous recombination, preventing the collapse of the DNA replication, and controlling other processes linked to RNA transcription and processing [23,24,25,26,27]. Thus, BRCA1-deficient cells show severe genomic instability characterized by chromosomal translocations, abnormal rearrangements, and a sensitivity to genotoxic agents. The role of BRCA1 as the guardian of genome stability is thought to account for the tumor-suppressive function of BRCA1 and some of its partners (i.e., BRCA2). Indeed, the loss of function mutations of BRCA1 are recurrent in familiar and sporadic breast and ovarian tumors.

Considering that the molecular events leading to TAZ activation in tumors are ill-defined and since TAZ mutations or mutations of its known upstream regulators are relatively rare in cancer, we set out to identify novel regulators of TAZ activity. We report here a genetic screen designed to identify regulators of TAZ transcriptional activity in mammary epithelial cells and describe an initial study of the potential regulation of TAZ by BRCA1 and other components of the BRCA complex.

## 2. Materials and Methods

### 2.1. Cell Lines and Culture Conditions

MCF10A cells were grown in Dulbecco’s Modified Eagle’s Medium (DMEM) mixed 1:1 with Ham’s F12K medium, supplemented with 5% Horse Serum, L-Glutamine (2 mM), human EGF (Epithelial Growth Factor) (20 ng/mL), Cholera Toxin (50 ng/mL), insulin (10 μg/mL), hydrocortisone (0.5 μg/mL), and 1% Penicillin-Streptomycin. The MCF10A–pSLIK–TAZS89A–8xTEAD–LUC cell line was cultured with 1.5 μg/mL of Puromycin and 400 μg/mL of Neomycin. The pSLIK–TAZS89A transgene was induced with 2 µg/mL of doxycycline for at least 48 h. The MCF10A–Cas9–sgRNA cell lines that constitutively express the Cas9 were grown with 1 μg/mL of Puromycin and 300 μg/mL of Neomycin. The MCF10A–rtTA–Cas9–sgRNA cell lines were grown with 1 μg/mL of Puromycin, 5 μg/mL of Blasticidin, and 300 μg/mL of Neomycin. Cas9 expression was induced by 1 µg/mL of doxycycline, and the KO efficiency was evaluated at 3 days or 8 days post-Cas9 induction. Cells were cultured in tetracycline-free serum to avoid spurious expression of the Cas9 transgene. The MCF10A cell lines carrying both inducible and constitutive shRNAs were selected and kept in culture with 1 μg/mL of Puromycin. For the inducible ones, the medium with tetracycline-free serum was supplemented with 1 µg/mL of doxycycline to switch on the expression of the short hairpin. The KD of the protein was evaluated after 3 days of shRNA induction.

MDA–MB–436 cells were grown in 90% DMEM mixed 1:1 with Ham’s F12 medium supplemented with 10% Fetal Bovine Serum (FBS, South America origin), L-Glutamine (2 mM), and 1% Penicillin–Streptomycin. HEK-293A cells WT or KO for LATS1, LATS2, or both LATS1/2 kinases were gifts from the Kun–Liang–Guan laboratory [28]. HEK–293T, HEK–293A, HeLa, and MDA–MB–231 cells were grown in 90% DMEM medium supplemented with L–Glutamine (2 mM), 1% Penicillin–Streptomycin, and 10% FBS (South America origin for HeLa, HEK–293A, and HEK–293A, while it is the North America origin for MDA–MB–231). All cell lines were grown in adhesion at 37 °C in 5% CO_2_. All the cell lines were tested and resulted negative for Mycoplasma. The proteasome was inhibited for 6 h with 5 µM MG132 (Sigma–Aldrich M7449, Merk Life Science S.r.l., Milano, Italy), while protein translation was inhibited with 100 µM of Cycloheximide (Sigma–Aldrich C4859, Merk Life Science S.r.l., Italy) for 2 to 4 h. 

### 2.2. Transfection

For the reverse transfection of short interference RNAs (siRNAs) in six well plates, the lipofectamine RNAiMAX reagent (Thermo Fisher Scientific, Life Technologies, Milan, Italy) was firstly diluted in 500 µL of Opti–MEM (Gibco, Life Technologies, Monza, Italy) and then incubated for 15 min at room temperature with 12.5 nM of siRNA, previously diluted in 500 µL of Opti–MEM to allow the assembling of liposomes complexes following the manufacturer’s instructions (1 mL of transfection mix per well). Cells were resuspended in medium with 2× FBS serum and without Penicillin–Streptomycin antibiotics. Afterward, cells were seeded in each well on top of the 1 mL of transfection mix at the following concentrations: 150.000 cells/well for MCF10A, 200.000 cells/well for HeLa and MDA–MB–231, 800.000 cells/well for HEK–293T and HEK–293A in a final volume of 2 mL per well (1 mL of cells and 1 mL of transfection mix). The KD was evaluated in sub-confluent cells after 48 h from the transfection. A list of siRNAs used with their relative sequences is reported in Appendix A.

For the transfection of plasmids, cells were previously seeded in six well plates, or in 10 cm plates in the absence of Penicillin–Streptomycin antibiotics, in order to reach 80% of confluence on the day of transfection. Following the manufacturer’s instructions (Thermo Fisher Scientific, Life Technologies, Italy), Lipofectamine3000 was diluted in an appropriate volume of Opti–MEM (Gibco, 125 µL for six well plates and 500 µL for 10 cm plates) and mixed 1:1 with a solution containing the DNA plasmids (2 µg per well of a 6well plate, 10 µg per 10 cm plate) and the p3000 Reagent (2 µL per µg of plasmid) previously diluted in Opti–MEM (125 µL for six well plates and 500 µL for 10 cm plates). After 15 min of incubation at room temperature, the transfection mix (250 µL for six well plates and 1 mL for 10 cm plates) was applied drop by drop over the sub-confluent adherent cells. After 6 h from the transfection, the medium was replaced with fresh, complete DMEM to eliminate the transfection reagents. The overexpression of the protein was evaluated after 24 h from the transfection. A list of plasmids used to transiently overexpress BRCA1, TAZ, LATS, and Ubiquitin proteins is reported in Appendix A. 

### 2.3. Viral Production and Cells’ Transduction

HEK–293T packaging cells were transfected with Lipofectamine 3000 following the manufacturer’s instructions (Thermo Fisher Scientific, Life Technologies, Italy) as described above. Briefly, for each 10 cm plate, 10 µg of the lentiviral plasmid of interest were mixed with helper plasmids (5 µg of VSVG and 6 µg of δ8.2) carrying the viral genes Pol, Env, and Gag. The transfection mix was applied drop by drop over adherent packaging cells for 6 h, and then the cell medium was replaced with fresh one (8 mL per plate). The viral supernatant was collected 24 h and 48 h post-wash out, filtered with 0.45 μm filter, and stored at −80 °C.

Target cells were incubated with 2 mL of lentiviral supernatant supplemented with 8 µg/mL of polybrene. After 8 h from the transduction, fresh medium was added to the cells to remove the polybrene, and after 24 h, cells were trypsinized and split into antibiotic-containing medium. All the plasmids used for viral production are listed in Appendix A with their relative references.

### 2.4. Cloning

All the sgRNAs against TAZ were designed with the CRISpick tool (https://portals.broadinstitute.org/gppx/crispick/public) and the sequences are reported in Appendix A. For cloning, the DNA oligonucleotides coding for the forward and reverse strands’ ends of the sgRNAs were phosphorylated in the presence of 5 U of T4 Polynucleotide Kinase enzyme (New England BioLab #M0201S, Euroclone, Pero, Italy) for 30 min at 37 °C, denatured at 95 °C for 5 min, and annealed by ramping down the temperature to 25 °C at 0.1 °C/s. For the ligation, annealed oligo sgRNAs were diluted 1:200 in water. Then, 5 µg of pLenti–sgETN backbone were digested with 20 U of the Bmsb I restriction enzyme (New England BioLab #R0580, Euroclone, Italy) for 3 h at 37 °C. Then, the digested plasmid was run on a 1% agarose gel to isolate the linearized backbone, which was purified from the gel following the QIAquick^®^ PCR and Gel Cleanup kit protocol (Qiagen S.r.l. #28506, Milano, Italy). After quantification by nanodrop, 1 µg of digested plasmid was dephosphorylated at 37 °C for 30 min in the presence of 1 U of rAPid Alkaline Phosphatase according to the manufacturer’s instructions (Sigma–Aldrich #4898117001, Merk Life Science S.r.l., Italy). Hence, we proceeded with the ligation step following the Roche protocol (Sigma–Aldrich #4898117001): 100 ng of the dephosphorylated vector were ligated to 2 µL of annealed sgRNAs (from the 1:200 dilution) for 30 min at room temperature in the presence of 5 U of T4 DNA Ligase. The ligation mix was transformed by heat shock in competent E.coli bacteria. The STBL3 bacterial strain was selected to reduce the frequency of homologous recombination of long terminal repeats proper of viral plasmids. Transformed bacteria were plated on plates with Luria Bertani medium supplemented with Ampicillin (50 μg/mL) and incubated at 37 °C overnight. The day after, three single colonies were picked and grown at 37 °C in 5 mL of Luria Bertani medium with Ampicillin for 8 h. For each bacterial culture, 4 out of 5 mL were used for plasmid extraction with the NucleoSpin^®^ Plasmid (No Lid) (Macherey-Nagel, Düren, Germany) according to the manufacturer’s instructions. The correct cloning of the sgRNAs was checked by DNA Sanger sequencing.

### 2.5. Immunoblot Analysis

For Immunoblot (IB) analysis, sub-confluent cells were lysed in six well plates by scraping cells on ice with 150 μL of cold lysis buffer (50 mM Tris–HCl pH 8, 300 mM NaCl, 1 mM EDTA, 1% NP40) supplemented with MINI-complete Protease Inhibitor and phospho–STOP inhibitor cocktail (Roche, Merck Life Science, Monza, Italy). Lysis was carried on for 20 min on ice. Then, cell lysate was sonicated with 10 cycles of 30 s ON/OFF at the maximum power (Bioraptor plus) and cleared by centrifugation at 13,000 round per minute (rpm) for 20 min at 4 °C. Extracted proteins were quantified by Bradford assay (Bio–Rad Laboratories S.r.l., Segrate, Italy). Then, 30 μg of proteins were boiled at 95 °C for 5 min with Laemli sample buffer (350 mM Tris–HCl pH 6.8, 30% glycerol, 10% SDS, 0.1% bromophenol blue, 6% β-mercaptoethanol) and loaded on 4–15% gradient precast TGXTM polyacrylamide gel (Bio–Rad Laboratories S.r.l., Italy). Proteins were blotted onto a nitrocellulose membrane using the Trans–Blot^®^ TurboTM transfer system for 30 min at 25 Volt and 1 Ampere. After the Ponceau staining and destaining, the membrane was firstly blocked with 5% of Bovine Serum Albumin (BSA) (*w*/*v*) diluted in Tris Buffered Saline supplemented with 0.1% (*v*/*v*) Tween (TBS-t) for 1 h at room temperature and then incubated overnight with the primary antibody at 4 °C. All the primary antibodies (listed in Appendix A) were diluted 1:1000 in TBS–t containing 5% of BSA. After three washes in TBS–t (15 min each), the membrane was incubated for 1 h at room temperature with the appropriate peroxidase-coupled secondary antibody diluted 1:10.000 in TBS–t. After 3 washes in TBS–t (15 min each), ECL-based chemiluminescence (Bio–Rad Laboratories S.r.l., Italy) was detected through the BioRad ChemiDoc system. Images were processed with Image Lab 4.0 (Bio–Rad Laboratories S.r.l., Italy).

### 2.6. TAZ Immunoprecipitation

For Immunoprecipitation (IP) experiments, sub-confluent HEK–293A cells, grown in 10 cm plates, were lysed by scraping cells on ice with 400 µL of cold protein lysis buffer (50 mM Tris–HCl pH 7.4, 150 mM NaCl, 1 mM EDTA pH 8, 0.5% NP40. 10% glycerol) supplemented with MINI-complete Protease Inhibitor, phospho–STOP inhibitor cocktail (Roche, Merck Life Science, Italy) and 2 mM PMSF. After 30 min of lysis on ice, cell lysate was cleared by centrifugation at 2500 rpm for 10 min at 4 °C. Extracted proteins were quantified by Bradford assay (Bio–Rad Laboratories). Then, 500 µg of whole cell extracts were incubated with 2 µg of primary antibody (listed in Appendix A) in a final volume of 500 mL of lysis buffer supplemented with 1 µg of BSA for 3 h on a rotating wheel at 4 °C. Then, 40 µL/IP of wet protein–A sepharose beads (for BRCA1 IP) or protein–G sepharose beads (for TAZ IP) were firstly blocked for 1 h with 2% BSA diluted in phosphate-buffered saline (PBS) and then added to cell extracts (previously mixed with the primary antibody) and incubated for 2 h on a rotating wheel at 4 °C. Bead–antibody–TAZ complexes were collected by centrifugation at 1000 rpm for 2 min at 4 °C and washed three times with 1 mL of cold Lysis Buffer by gently inverting the tube 5–10 times. To remove the residual amount of lysis buffer, at the end of the third wash, a syringe was used to dry the beads. Finally, protein complexes were eluted in 30 µL of 2× Laemli sample buffer (350 mM Tris–HCl pH 6.8, 30% glycerol, 10% SDS, 0.1% bromophenol blue, 6% β-mercaptoethanol), boiled for 5 min at 95 °C, loaded on 4–15% gradient precast TGXTM polyacrylamide gel (Bio-Rad Laboratories) and blotted on nitrocellulose membranes through the Trans–Blot^®^ TurboTM transfer system for 30 min at 25 Volt and 1 Ampere. After 1 h of blocking with 10% BSA diluted in TBS–t, the membrane was incubated with the primary antibody overnight at 4 °C and the day after with the appropriate peroxidase-coupled secondary antibody for 1 h at room temperature. Dilution of antibodies, washes, and chemiluminescence detection were performed as described for IB analysis. 

For ubiquitination studies, the protein lysis buffer was supplemented with 25 µM of PR619 to preserve ubiquitinated proteins from endogenous DUBs and sepharose beads were crosslinked with TAZ antibody to avoid the detection of the heavy chains of the primary antibody at 55 kDa in the proximity of TAZ band. For the crosslinking, 50 µL of wet beads per IP were previously washed two times at room temperature with 1 mL of 0.2 M Na_2_B_4_O_7_ solution pH 9, collected by centrifugation at 2000 rpm for 2 min at room temperature, then resuspended in Na_2_B_4_O_7_ solution (two times the initial volume) and incubated with 1 µg of TAZ antibody for 1 h on a rotating wheel at room temperature. After three washes with 1 mL of 0.2 M Na_2_B_4_O_7_ solution pH 9, the bead–antibody complexes were resuspended in 1 mL of dimethyl pimelimidate (DMP) solution (the powder, Thermo Fisher # 21667, Life Technologies, Italy, was resuspended in 0.2 M Na_2_B_4_O_7_ solution pH 9 in order to reach a final concentration of 5 mg/mL) and incubated for 45 min on a rotating wheel at room temperature. To quench the reaction, the beads covalently bound to the TAZ antibody were washed twice with 1 mL of 0.2 M ethanolamine pH 8.0 and successively incubated with 1 mL of 0.2 M ethanolamine pH 8 for 2 h on a rotating wheel at room temperature. After three washes in PBS, bead–antibody complexes were used for immunoprecipitating TAZ. Before proceeding with TAZ IP, cell extract was precleared to eliminate any aspecific binding: 2 mg of cell lysate were incubated with 50 µL of wet sepharose beads for 3 h on a rotating wheel at 4 °C. Then, 40 µL of bead–antibody complexes were incubated overnight with 2 mg of precleared cell extract on a rotating wheel at 4 °C to perform TAZ immunoprecipitation.

### 2.7. RNA Extraction and RT–qPCR

For expression analysis, sub-confluent cells were trypsinized, centrifugated at 1200 rpm for 5 min, and washed once in PBS. RNA was extracted by using the Quick–RNA MiniPrep kit (Zymo Research, Euroclone, Pero, Italy) in accordance with the manufacturer’s instructions. Briefly, cells were lysed by adding 300 µL of RNA lysis buffer to the cells pellet, and the lysate was passed through a Spin–Away™ filter to remove the majority of genomic DNA. The flowthrough was mixed 1:1 with ethanol 100% and loaded onto a Zymo–Spin™ IIICG Column to capture RNA. The column was washed with 400 µL of RNA Wash Buffer before proceeding with the DNAseI treatment: 5 µL of DNase I (1 U/µL) were mixed with 75 µL of DNA Digestion Buffer, and the mixture was applied directly onto the column matrix. After 15 min of incubation at room temperature, the column was washed first with 400 µL of RNA Prep Buffer and then twice with 700 µL of RNA Wash Buffer. An additional centrifugation at full speed for 1 min at room temperature was performed to eliminate residual buffers. RNA was eluted by adding 30–50 μL of RNase-free water onto the column and centrifuging for 1 min at full speed. The extracted RNA was quantified using Nanodrop (Thermo Fisher). 

Purified RNA was then used for cDNA synthesis with ImPromII Reverse Transcription System (Promega, Milano, Italy). Then, 1µg of RNA was mixed with 1 μL of Oligo–dT and 0.1 μL of random primers diluted in RNase-free water in a final volume of 30.25 μL and incubated for 5 min at 70 °C to eliminate secondary structures. After 5 min on ice, samples were mixed with 19.75 μL of an RT solution containing 5 μL of 25 mM MgCl2, 10 μL of 5X RT buffer, 2.5 μL of 10 mM dNTP mix, 1.25 μL of RNase inhibitor, and 1 μL of Reverse Transcriptase (RT). The mixture was incubated first at 25 °C for 5 min to allow primers annealing, then at 42 °C for 60 min to enable RT elongation, and lastly at 70 °C for 15 min to inactivate the RT enzyme.

Synthesized cDNA was used for subsequent Real-time quantitative PCR (RT–qPCR). First, 10 ng of cDNA were mixed with 10 μL of Sybr Green 2X (Thermo Fisher) and with 500 nM of primers in a final reaction volume of 20 μL. RT–qPCR reactions were performed with BioRad CFX 96 System in three technical replicates. The data were analyzed by using the 2^–ΔΔCt^ method. Primers’ sequences are reported in Appendix A.

### 2.8. Proximity Ligation Assay and Immunofluorescence Analyses

For Immunofluorescence (IF), staining cells were seeded onto glass coverslips. Sub-confluent cells were fixed with 4% paraformaldehyde (PFA) diluted in PBS for 15 min at room temperature. After three washes in PBS, fixed cells were permeabilized with 0.2% Triton X-100 (diluted in PBS) for 15 min at room temperature. Aspecific signals were reduced by blocking cells with 1 mL of blocking solution (2% BSA in PBS) for 1 h at room temperature. Then, glass coverslips were incubated for 1 h at room temperature with primary antibodies (anti-TAZ (C22) and anti-BRCA1 Bethyl, listed in Appendix A) diluted 1:200 in blocking solution. Cells were washed three times with PBS before being incubated for 1 h at room temperature with fluorophore-conjugated secondary antibodies diluted 1:500 in blocking solution, light-protected. After three washes in PBS, nuclei were stained with DAPI (diluted 1:5000 in PBS) for 5 min at room temperature, and glass coverslips were mounted on slides with mowiol or glycerol (the latter for co-localization experiments). For Proximity Ligation Assay (PLA), cells were seeded onto glass coverslips, and when sub-confluent, cells were fixed with 4% paraformaldehyde (PFA) diluted in PBS for 15 min at room temperature. Fixed cells were permeabilized with 0.2% Triton X-100 (diluted in PBS) for 15 min at room temperature and then blocked with Duolink Blocking Solution for 1 h at 37 °C. Afterward, cells were processed following the Duolink In Situ PLA protocol. All the incubation steps were performed in a pre-heated humidity chamber, and from the amplification step, slides were protected from light. For PLA analysis, cells were imaged on the Leica Multi-fluorescent wide-field microscope (DM6 B) for IF and PLA assay, while on the Leica confocal wide-field SP8 microscope, DMi8 (inverted) for co-localization assay. Cells were recorded by using the oil-immersion 63X objective (for co-localization assay) (NA 1.40) or the 100X objective (for PLA and IF assay) (NA 1.40). For PLA analysis, we acquired 15–20 Z-stacks of 0.1 μm thickness for each field of view. The analysis of the images was performed with Fiji software (version 2.14.0/1.54f) by implementing a custom-made macro that allowed to automate the analysis. Briefly, the PLA dots coming from different stacks of the same cell were projected in a 2D space by applying the maximum intensity Z projection method. Cell area was measured by an ImageJ plugin that predicts polygonal shapes around the DAPI signal. PLA dots were counted with the Find Maxima Fiji function. Concerning the measurement of TAZ mean intensity, the analysis of the images was performed with Fiji software through the implementation of a custom-made macro that automatically assesses TAZ nuclear mean intensity, which signal overlaps with the Dapi and TAZ cytosolic mean intensity, which signal was measured in a perinuclear crown.

### 2.9. siRNA Screening

The esiRNA custom library was purchased from Sigma (MISSION^®^ esiRNA, Sigma) in 384-well format. esiRNAs are endoribonuclease-prepared siRNA pools comprised of a mixture of siRNAs that all target the same gene. Control esiRNAs targeting human PLK1 (essential gene, positive control for transfection), ATM (used as a neutral siRNA), STK3 (repressor of TAZ), and TAZ/WWTR1 were purchased in individual tubes and included in the final 384-well plates. The screen was performed by retro-transfecting MCF10A–pSLIK–TAZS89A–8xTEAD–LUC cells. In brief, each well of a 348 well plate was filled with 113µL of Optimen (GIBCO, Life Technologies, Italy), 0.7 µL of RNAiMAX (ThermoFicher Scientific, Life Technologies, Italy), and 7 µL of 1 µM siRNA solution. The mix was incubated for 30′ at room temperature and then aliquoted in six 384 wells (20 µL/well). One thousand cells (resuspended in 20µL) were added to each well. A total of six replicates plates (three mock- and three treated with 1µg/mL Doxycycline) were transfected for each of the library siRNAs. Cells were grown for 48 h in standard conditions and then processed. Viability was assessed by Real-Time Glo (Promega, Italy), while TEAD reported activity by One Glo (Promega, Italy). Luminescence values for both viability and TEAD reporter activity were Z-score normalized. Robustness of positive hits was evaluated by calculating the strictly standardized mean difference (SSMD) for TEAD reporter values. SSMD is the mean of log fold change divided by the standard deviation of log fold change with respect to a negative reference (i.e., neutral siRNAs). Ranking of hits was based on SSMD values, using the standard setting: 5 > |SSMD| ≥ 3 for very strong, 3 > |SSMD| ≥ 2 for strong candidates [29].

### 2.10. Statistical Analyses

For samples up to *n* = 3, statistical significance was assessed by using the two tailed Student *T*–test. Instead, when sample size was higher, statistical significance was assessed by the Mann–Whitney test. Analyses were performed with the PRISM software (GraphPad, version 9.4.1). 

## 3. Results

### 3.1. A Genetic Screen to Identify Regulators of TAZ Activity

To identify regulators of TAZ activity, we devised a loss of function genetic screen in a TAZ-reporter cell line (MCF10A–pSLIK–TAZS89A–8xTEAD–LUC) (Figure 1a). This cell line expresses (i) A luciferase reporter with eight repeats of the TEAD-binding motifs and (ii) A doxycycline-regulated TAZ mutant (i.e., TAZS89A, activated mutant). We designed a custom siRNA library of 1600 genes (Appendix A), which were selected for being regulated by TAZ in MCF10A cells, with the rationale that TAZ targets often participate in feedforward and feedback regulatory loops, as emerged by transcriptional analysis performed in our lab and as reported by others [30,31,32,33]. The library also contained neutral siRNAs, siRNAs targeting essential genes, and siRNAs targeting known regulators of YAP/TAZ activity, which were used to assess the reliability and sensitivity of the viability and TAZ-reporter assay (Appendix A) and for setting the thresholds for hit calling during the analysis of the screen (Figure 1b,c). The screening was performed in biological triplicates in 384 well plates, with automatic liquid handling to reduce the operator-based variability and to increase reproducibility. For each condition, cells were seeded in technical replicates in the presence of doxycycline to induce the expression of TAZS89A or without doxycycline to evaluate the modulation of the endogenous TAZ and YAP. Upon seeding, cells were transfected with siRNAs (each well with a siRNA-mix targeting a unique transcript). After 48 h, cell viability and TAZ transcriptional activity were evaluated by bioluminescence assays (Figure 1a).

Ectopic expression of TAZS89A led to a pronounced activation of the TEAD reporter (Appendix A). Of note, the KD of TAZ significantly reduced reported activity, while MST2/STK3 loss enhanced it (Appendix A). However, we detected mild variations of the TAZ reporter when probing the activity of the endogenous TAZ, thus suggesting that this system was unsuitable to probe endogenous TAZ (Appendix A). For analysis, viability and TEAD–reporter activity for each targeted gene were Z-score normalized. For hit calling, viability and reporter activity thresholds were set based on the values obtained when silencing PLK1 (essential gene used for setting the viability threshold), STK3/MST2, and WWTR1/TAZ (used for setting the reported activity threshold). Overall, we identified around 200 genes whose silencing increased (i.e., TAZ inhibitors) or decreased (i.e., TAZ activators) TAZ transcriptional activity (Figure 1b–d, Appendix A). By this screen, we confirmed some of the known regulators of YAP/TAZ, such as WTIP [34], BRD4 [35], and members of the WNT-signaling pathway [8] (FZD2, WNT10B, FZD5, TCF7) (Appendix A). GO analyses by Metascape (https://metascape.org/gp/index.html#/main/step1) highlighted a prevalence of hits involved in cytoskeletal regulation, cell junction, and motility (Figure 1e and Appendix A). We also identified a consistent fraction of genes involved in interleukins and NF-kappaB signaling (as EOMES, GFPT2, SOCS2, P2RX4, GMPR, STAT5A, PTGER2, CSF1, SLC2A6), possibly suggesting a cross-talk of TAZ activity and inflammatory pathways (Figure 1e and Appendix A). In total, we identified 91 genes that, when silenced, led to higher reported activity (UP-hits, repressors of TAZ), 82 genes that, when silenced, led to lower TAZ-reporter activity (down-hits, activators of TAZ), and 43 genes that once silenced led to both lower TAZ-reporter activity and lower viability (down-lethal hits) (Figure 1d and Appendix A). 

### 3.2. BRCC3 and the BRCA1–A Complex Regulates TAZ

Among the TAZ inhibitors, BRCC3 emerged as one of the most interesting hits (Figure 1). BRCC3 is a deubiquitinating enzyme (DUB) belonging to the JAMM/MPN+ family of zinc metalloproteases that regulates the abundance of Lys-63 (K63) polyubiquitin chains both in chromatin, as well as in other cytosolic substrates involved in stress response and immune-signaling functions [36,37,38,39]. Interestingly, BRCC3 can participate in two different multiprotein complexes: the BRCA1-A complex, which safeguards genome integrity by regulating DNA repair, and the BRISC complex, which is involved in diverse cellular functions depending on the specificity of the bound substrate [40]. 

Next, to confirm the identification of BRCC3 as a repressor of TAZ activity, we assessed whether silencing BRCC3 would modulate the activity of TAZ, both the endogenous one and the ectopically expressed TAZS89A. We monitored the expression of CTGF and Cyr61, two well-established TAZ-target genes. At the endogenous level of TAZ (no doxycycline treatment), BRCC3 KD raised the expression of Cyr61, while upon TAZS89A expression, both TAZ target genes are significantly upregulated (Figure 2a), thus confirming the result of the screen.

TAZ activity is mainly regulated by post-translational modifications, which affect its protein stability and its cellular localization [2,41]. Therefore, we proceeded by investigating whether the observed gain in TAZ activity might correspond to an increase in the total amount of TAZ. WB analysis revealed that upon silencing of BRCC3, there was a consistent increase of TAZ protein, observed for both the endogenous TAZ and the ectopically expressed TAZS89A (Figure 2b). These results suggest that BRCC3 is a repressor of TAZ, able to control both TAZ activity and TAZ protein amount. Additional experiments on BRCC3 are ongoing and will be documented in a separate report. Next, we focused on the BRCA1–A complex, in which the activity of the deubiquitinating enzyme BRCC3 is balanced by the ubiquitinating enzyme BRCA1 [42]. In order to evaluate whether this complex can regulate TAZ activity, we silenced transcripts encoding some of the critical subunits of the BRCA1–A complex: BRCA1, BRCA2, and BARD1. In all the cases, we achieved a successful silencing that was evaluated either by WB (BRCA1, Figure 2g) or RT–qPCR (BRCA2 and BARD1, Figure 2c,d). Loss of BRCA2, BARD1, or BRCA1 expression raised TAZ protein levels (Figure 2d,e), thus implying that the BRCA1–A complex may regulate TAZ. This suggested that the activity of the BRCA1–A complex relies on a stoichiometric balance among its components, therefore perturbing this balance may disrupt the activity of the complex and the regulation of TAZ protein.

Due to the relevance of BRCA1 as a tumor suppressor in basal-like breast cancers (BLBCs) and considering that these tumors are also characterized by high levels of TAZ activity, we focused on BRCA1. 

To generalize our observations, we silenced BRCA1 in MCF10A–pSLIK–TAZS89A–8xTEAD–LUC, MDA–MB–231, HeLa, and HEK–293T cells loss, and assessed the TAZ level by WB. In all the cases, we consistently detected a significant increase in TAZ protein levels upon BRCA1 loss (Figure 2f,g). Notably, in MCF10A–pSLIK–TAZS89A–8xTEAD–LUC, the regulation of TAZ abundance was observed also upon the expression of the exogenous TAZS89A (Figure 2f), implying post-transcriptional regulation of TAZ by BRCA1.

Differently from what was observed for TAZ, we did not note a consistent pattern of modulation of YAP once BRCA1 was silenced. Indeed, YAP increased in MCF10A–pSLIK–TAZS89A–8xTEAD–LUC and MDA–MB–231, it slightly decreased in HEK–293T, while it remained stable in HeLa (Appendix A).

Next, we asked whether the increment in TAZ protein levels observed upon BRCA1 loss would lead to an increase in TAZ activity. BRCA1 silencing in MCF10A–pSLIK–TAZS89A–8xTEAD–LUC significantly increases the expression of CTGF and Cyr61 both in the absence of doxycycline treatment and upon expression of the ectopic TAZS89A (Figure 2h).

To extend these observations, BRCA1 was silenced in other cell lines: MDA–MB–231, HeLa, and HEK–293T cells (Figure 2i). In all the cases, TAZ transcriptional activity was raised upon BRCA1 silencing, thus supporting the hypothesis of a general regulation of TAZ by BRCA1, both in normal and in tumor cells.

### 3.3. BRCA1 Does Not Control TAZ Cellular Localization

Since TAZ activity depends on its nuclear localization as well as its protein stability, we asked whether BRCA1 silencing would alter the cellular localization of TAZ by performing IF analyses with two TAZ-specific antibodies (C22 and C188). Coherently with the WB results, both antibodies revealed a significant gain in the relative intensity of TAZ signal in the nuclear compartment as well as in the cytosolic one upon BRCA1 silencing (Figure 3b). Despite this general gain, we did not observe any variation in TAZ sub-cellular distribution (Figure 3a) since TAZ increased both in the cytoplasm and in the nucleus, thus suggesting that BRCA1 regulates TAZ levels without unbalancing its nuclear–cytosolic ratio.

### 3.4. BRCA1 May Regulate TAZ in the Cytoplasm

To seek evidence of direct regulation by BRCA1, we evaluated TAZ–BRCA1 spatial proximity by Proximity Ligation Assay (PLA). We were able to observe PLA–foci, which suggests proximity between endogenous TAZ and BRCA1 in HeLa (EV) (Figure 4a). To evaluate the specificity of these signals, we transiently overexpressed both the target proteins. In the presence of the ectopic proteins, we detected an increased number of PLA–foci per cell (i.e., dots/cell) (Figure 4a), thus confirming that proximity depends on the relative abundance of the two proteins. As negative technical controls for the PLA, we incubated cells with only one primary antibody (either anti-TAZ or anti-BRCA1) or without primary antibodies (only secondary probes), and we observed a drop in the signals to background level. To reinforce these observations, we performed PLA experiments in MCF10A cells and in isogenic MCF10A TAZ–KO lines, derived by CRISPR-Cas9 genomic editing (Appendix A). Consistently with the previous experiments, we detected PLA signals indicating intracellular physical interaction of BRCA1 and TAZ; these foci were markedly reduced in the TAZ CRISPR–KO cells, thus confirming the specificity of the PLA signal (Figure 4b,c). Overall, in two independent cellular models, the HeLa and the MCF10A cells, we consistently observed a specific molecular proximity between BRCA1 and TAZ proteins. We measured the relative fraction of the nuclear and cytosolic PLA spots in HeLa and MCF10A cells overexpressing TAZ and BRCA1 proteins. To identify the two compartments, we took advantage of the DAPI as nuclear marker and of the ectopic TAZ signal, which stained all the transfected cells with a diffuse pattern, thus limiting cell borders. In both cellular models, the PLA spots are preferentially distributed in the cytosolic compartment (Figure 4d,e).

### 3.5. LATS1/2 Are Epistatic over BRCA1 in Regulating TAZ Levels

The Hippo pathway is a central upstream regulator of TAZ activity. Therefore, we asked whether BRCA1 might regulate TAZ by affecting components of this pathway. 

To test this, we performed WB analysis upon BRCA1 knock-down (BRCA1–KD) (Figure 5a,b). While MST2 and LATS2 proteins’ levels were not affected by BRCA1 loss, LATS1 and its partner MOB1 (only in the MCF10A cell line) decreased, thus suggesting that BRCA1 may regulate some Hippo components. This result may indicate a possible involvement of LATS kinases in the BRCA1-mediated regulation of TAZ. While interesting, these alterations were unlikely to account for the observed selective modulation of TAZ since LATS1,2 are not haploinsufficient for the regulation of YAP/TAZ. 

Nonetheless, to test more directly the role of LATS1 and 2 in the regulation of TAZ by BRCA1, we took advantage of HEK–293A cell line KO for LATS1, LATS2, or both (dKO) and asked whether LATS1,2 deficiency would prevent regulation of TAZ by BRCA1. In single LATS–KO cells, BRCA1 silencing still raised TAZ levels (Figure 5c). In addition, single KO cells did not show alterations in TAZ levels, thus confirming that the expression of either LATS1 or LATS2 is sufficient to keep TAZ under Hippo regulation. As reported, TAZ levels increased in dKO cells. However, in these cells, the silencing of BRCA1 did not lead to a further increase of TAZ (Figure 5c). 

With the aim of corroborating the above data, we proceed by restoring LATS1 protein levels in the dKO cell line to rescue LATS1 activity concomitantly to the transient KD of BRCA1. As expected, the ectopic expression of LATS1 in the dKO cell line reduced TAZ levels (Figure 5d). In these cells, the additional silencing of BRCA1 led to an increase in TAZ abundance, thus indicating that restoration of LATS1 expression was sufficient to rescue BRCA1-mediated regulation of TAZ in the dKO cell line (Figure 5d). These findings imply that BRCA1-mediated regulation depends on LATS1/2 kinases, thus suggesting an epistatic role of LATS1/2 kinases over BRCA1. These results indicate that BRCA1 requires a Hippo signaling to modulate TAZ activity.

### 3.6. BRCA1 Controls TAZ Expression Mainly by Post-Translational Mechanisms

The results collected so far mostly fit with the hypothesis of a post-translational mechanism of regulation. To test this, we investigated whether BRCA1–KD would affect TAZ transcription, translation, or protein degradation.

Firstly, we checked whether BRCA1 might modulate TAZ transcription and noted that TAZ mRNA was consistently upregulated upon BRCA1–KD in multiple cell lines (Figure 6a,b), thus suggesting that BRCA1 represses TAZ transcription. The same transcriptional regulation was confirmed in HEK–293A cell lines, wild type, or LATS–dKO (Figure 6c), thus indicating that LATS kinases are unlikely to be involved. Notably, in the LATS1/2 dKO cell line, TAZ was still transcriptionally modulated upon BRCA1–KD despite its unchanged protein levels, thus indicating that the transcriptional upregulation does not necessarily contribute to the overall gain in TAZ protein, and also suggesting that BRCA1 may regulate TAZ post-transcriptionally. To further assess this, and considering that TAZ is subjected to steady-state proteasomal degradation, we blocked TAZ degradation using the proteasome inhibitor MG132 and assessed whether BRCA1 silencing would affect TAZ protein synthesis. We reasoned that under these conditions, the total amount of TAZ would depend on the levels of TAZ transcription and translation (i.e., TAZ synthesis). We performed this experiment in confluent MCF10A cells, a condition that favors TAZ proteasomal degradation. As reported, when the proteasome was blocked, TAZ protein increased, thus confirming that proteasomal degradation limits TAZ cellular levels (Figure 6d). However, the gain observed upon the MG132 treatment was comparable in the presence of BRCA1, as well as upon BRCA1–KD, thus suggesting that BRCA1 does not affect TAZ synthesis (Figure 6e). In addition, this also suggests that the gain in TAZ mRNA level consistently observed upon BRCA1 silencing cannot account for the increase in TAZ protein since if the increased RNA level was the main reason for the enhanced TAZ abundance, the rise in TAZ protein would be still observed upon proteasome inhibition.

### 3.7. BRCA1 Regulates TAZ Poly-Ubiquitination

The previous results indicated that TAZ synthesis is not regulated by BRCA1, thus suggesting that BRCA1 might regulate TAZ by controlling its degradation (at the protein level). To address this point, we blocked the translation with Cycloheximide (CHX) (thus blocking protein synthesis), and we assessed whether BRCA1 silencing would affect TAZ levels by modulating its degradation. Indeed, in the absence of de-novo protein synthesis, TAZ abundance would depend only on its stability/degradation. Coherently with the previous experiments, we chose to work with confluent MCF10A cells to maximize the Hippo-mediated TAZ turnover. In wild-type cells (siC), we observed a progressive decrease in the amount of TAZ, with a significant drop at 2 h, in line with the reported TAZ half-life (Figure 7a). Differently, TAZ degradation was reduced upon BRCA1-KD (Figure 7a,b), thus suggesting BRCA1 may negatively regulate TAZ stability.

Overall, the results collected so far supported a link between the regulation of TAZ stability and BRCA1. Due to the ubiquitin ligase activity of BRCA1, we investigated whether TAZ stability might depend on BRCA1-mediated ubiquitination. To examine this hypothesis, we immunoprecipitated TAZ and looked for its ubiquitination upon BRCA1–KD (Figure 7d). We overexpressed TAZ as well as Ubiquitin with the purpose of increasing the amount of TAZ that can be degraded and fostering its ubiquitylation. Additionally, we inhibited the proteasome with MG132 to limit TAZ turnover and enrich the lysate for polyubiquitinated TAZ proteins. In all the experimental conditions, we achieved an efficient immunoprecipitation (IP) of TAZ, which was enhanced upon TAZ overexpression (Figure 7c). Despite the weak signals in endogenous condition (EV), we were able to confirm that TAZ ubiquitination was reduced upon BRCA1–KD (Figure 7d). The overexpression of both TAZ and Ubiquitin (TAZ+Ub) enhanced the abundance of poly-ubiquitinated TAZ species, which decreased in the absence of BRCA1 (Figure 7d). As expected, the block of the proteasome in endogenous condition (EV) raised TAZ ubiquitination compared to the basal signal; by contrast, BRCA1–KD consistently reduced TAZ ubiquitination (Figure 7d). A similar regulation was observed upon MG132 treatment in the overexpression condition (TAZ+UB) (Figure 7d). In summary, the above results suggest that BRCA1 may regulate TAZ degradation by affecting its ubiquitination state (Figure 8).

## 4. Discussion

In this study, we show that the tumor suppressor BRCA1 can regulate TAZ activity by modulating its ubiquitination and protein stability via ubiquitin-dependent proteasomal degradation (Figure 7d). Epistatic analysis and loss of function studies indicated that BRCA1-mediated inhibition of TAZ required a competent Hippo-signaling pathway (Figure 5), thus suggesting integrated regulation of TAZ by both Hippo and BRCA1. The observation that upon silencing of BRCA1, LATS1 (and possibly MOB1) are consistently down-modulated may also suggest an additional layer of cross-talk between BRCA1 and the Hippo pathway, whereby BRCA1, either directly or indirectly may also modulate the activity of other components of the Hippo pathway. At this point, it is still an open question how BRCA1 may regulate LATS1 levels, yet this is unlikely to be sufficient for TAZ regulation, as LATS1-KO cells do not show upregulation of TAZ (Figure 5c), thus indicating that LATS1 loss is compensated by LATS2.

While the loss of BRCA1 affects also transcription of the TAZ mRNA, our data suggest that BRCA1 regulates TAZ mainly by controlling it ubiquitin mediated proteasomal degradation.

We also identified BRCC3, a lysine-63-specific deubiquitylating enzyme (DUB) and member of the Zn^2+^ dependent JAB1/MPN/Mov34 metalloenzyme (JAMM) domain metalloprotease family, as capable of regulating TAZ. BRCC3 is a component of the BRCA1 complex involved in in the resolution of the activity of the BRCA1 complex by catalyzing the cleavage of the K63-ubiquitin chains from H2A and H2AX, thus releasing the BRCA1 complex from chromatin [43,44] BRCC3 is also a member of the BRISC complex which regulates various pathways linked to inflammatory responses by regulating ubiquitylation of some components of the inflammasome and the interferon receptor IFNAR1 [36,39]. Whether BRCC3 regulates TAZ because of its activity within the BRISC or the BRCA1 complex will require further studies. Yet, considering that the loss of other components of the BRCA-1 complex phenocopy loss of BRCA-1 or BRCC3, it is conceivable that the activity and integrity of the BRCA1 complex are required for the proper regulation of TAZ. 

This may also explain how targeting either BRCA-1 (a ubiquitin ligase) or BRCC3 (a deubiquitylating enzyme), which support opposing biochemical reactions, may lead to stabilization of TAZ, because of the altered stoichiometry and functionality of the entire BRCA1 complex.

Despite YAP/TAZ share structural and functional homology and are most frequently coregulated by the same upstream pathways, our data point to selective regulation of TAZ, while YAP protein stability did not seem to be consistently regulated by BRCA1-silencing. Indeed, knock-down of BRCA1 in MCF10A and MDA-MB-231 cells led to YAP stabilization. In HEK-293T, we noted an opposite modulation, while YAP was not modulated in HeLa (Appendix A). Collectively, these observations may suggest that BRCA1 specifically regulates the TAZ protein. Yet, considering previous evidence reporting regulation of YAP activity by BRCA1, through modulation of NF2 stability [45], at this point, we cannot exclude that BRCA1 may also regulate YAP in a cell/context-dependent manner and through mechanisms that do not rely on the control of its stability [46,47]. 

Collectively, our findings highlight a novel genetic axis that, in perspective, may have a potential clinical relevance in cancer. Our data suggests that the loss of the BRCA1 tumor-suppressor gene may potentially drive the activation of the TAZ oncogene. 

Several studies highlighted a correlation between high levels of TAZ and the aggressive phenotype of basal-like, poorly differentiated, high-grade mammary tumors, which showed the worst response to pharmacological treatments among the breast tumor sub-types [21,22,48,49,50]. On the other hand, loss of function mutations of BRCA1, which often occur at the early stages of breast and ovarian tumors, have been associated to epithelial-to-mesenchymal transition and transcriptional reprogramming of luminal cells into basal cells [51,52,53,54]. The co-occurrence of BRCA1 loss and TAZ gain may represent a genetic feature of basal-like breast tumors and may provide a rationale for the aggressiveness of BRCA1 mutated tumors, where inactivation of the BRCA1 tumor suppressor may also have the effect of activating the TAZ oncogenic program. Overall, our study may have the potential to uncover a relevant link between the guardian of our genome (i.e., BRCA1) and a master regulator of cell proliferation (i.e., TAZ), which might represent an evolutionary response of cancer cells to the detrimental effects of the loss of genome integrity pathways.

## 5. Conclusions

This study identifies new regulators of TAZ activity and provides evidence that BRCA1 and other members of the BRCA1 complex regulate TAZ ubiquitylation and proteasomal degradation.

## Figures and Tables

**Figure 1 cells-12-02431-f001:**
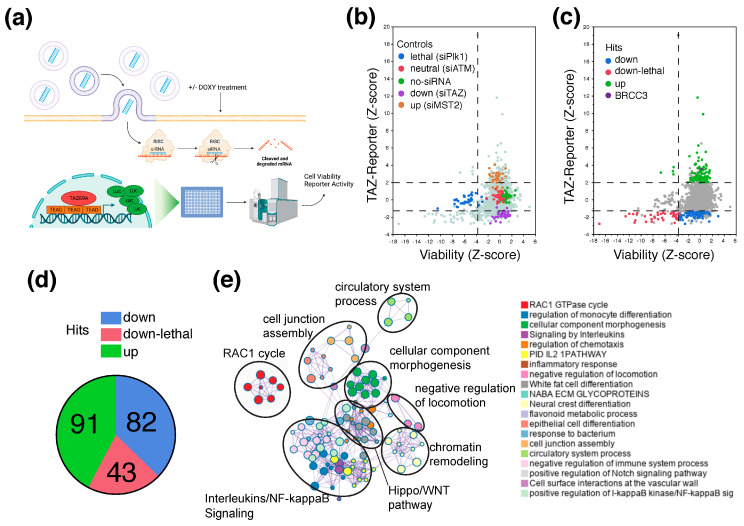
Identification of genes regulating TAZ activity by an siRNA screen. (**a**) Outline of the siRNA screen. (**b**,**c**) Dot plot of TEAD reporter activity and cell viability (Z-score normalized values). The different classes of control siRNA (**c**) and hits (**d**) are highlighted in colors. (**d**) Pie chart of the different classes of hits. (**e**) Gene Ontology network of the genes identified as regulators of TAZ activity.

**Figure 2 cells-12-02431-f002:**
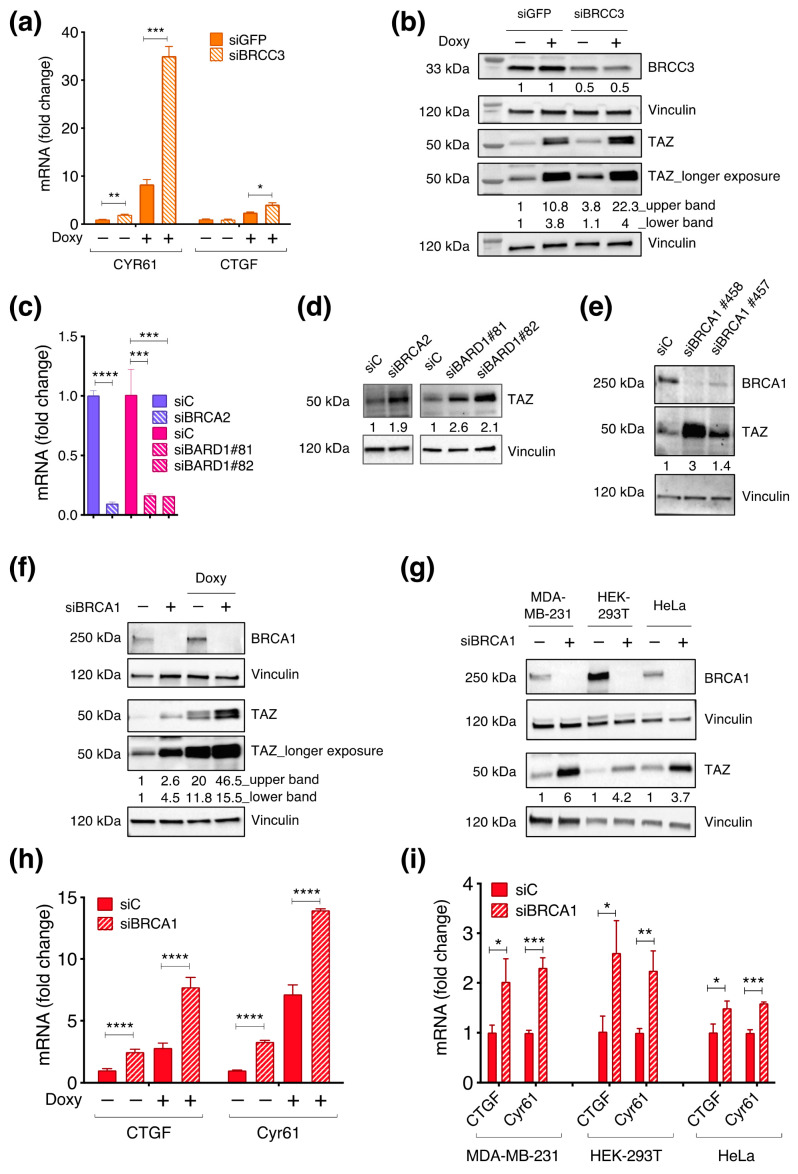
BRCC3, BRCA1, and other BRCA−1 complex components repress TAZ activity. (**a**,**b**) First, 150.000 pSLIK–TAZS89A–8xTEAD–LUC MCF10A cells were transfected with siRNA against BRCC3 or with a non−targeting sequence (siGFP) as control, in the presence or in absence of 2 µg/mL of doxycycline. After 48 h from transfection, cells were collected. (**a**) RT–qPCR analysis. The bar plot shows cDNA levels of two TAZ target genes, CTGF, and CYR61, normalized on GAPDH housekeeping gene and expressed as fold change. *T*–test was applied to evaluate the statistical significance: * *p* value < 0.05 ** *p* value < 0.01 *** *p* value < 0.005. (**b**) WB analysis, vinculin was used as loading control. (**c**,**d**) Silencing of BARD1 and BRCA2 in MCF10A cells. RT–qPCR and WB analysis were performed after 48 h from the transfection in sub-confluent condition. (**c**) Bar plots showing the expression levels of BRCA2 and BARD1. *T*–test was applied to evaluate the statistical significance: *** *p* value < 0.005 **** *p* value < 0.001. (**d**) WB analysis, vinculin was used as loading control. (**e**–**g**) Analysis of TAZ and BRCA1 protein level by WB, following BRCA1−KD in the reported cell lines. (**b**,**d**–**g**) The normalized densitometric values of the WB bands are reported below the WB snapshots. (**h**,**i**) Analysis of TAZ target genes by RT–qPCR in (**h**) pSLIK–TAZS89A–8xTEAD–LUC MCF10A cells and (**i**) MDA–MB–231, KEK–293T, and HeLa cells.

**Figure 3 cells-12-02431-f003:**
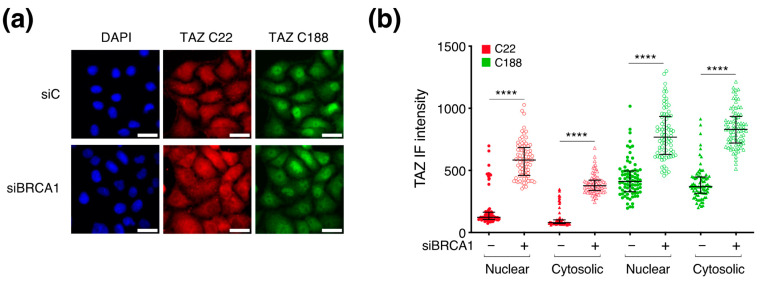
Silencing BRCA1 alters TAZ protein levels without affecting its intracellular distribution. For the evaluation of TAZ by Immunofluorescence analysis, HeLa cells were transfected with siBRCA1 (#458) or with a non−targeting siRNA, as control (siC). After 48 h, sub−confluent cells were fixed and stained with two TAZ antibodies. (**a**) Representative images at 100× magnification, scale bar = 30 µm, in blue Dapi stained nuclei, in red TAZ protein detected by the C22 antibody, in green TAZ protein stained with the C188 antibody. (**b**) Dot plot showing the quantification of the mean intensity of TAZ relative signal per cell detected with the C22 and the C188 antibody. Raw data were analyzed through the Fiji software (version 2.14.0/1.54f). Sample size: 80 cells. Mann−Whitney test was applied to run statistical analysis: **** *p* value < 0.0001.

**Figure 4 cells-12-02431-f004:**
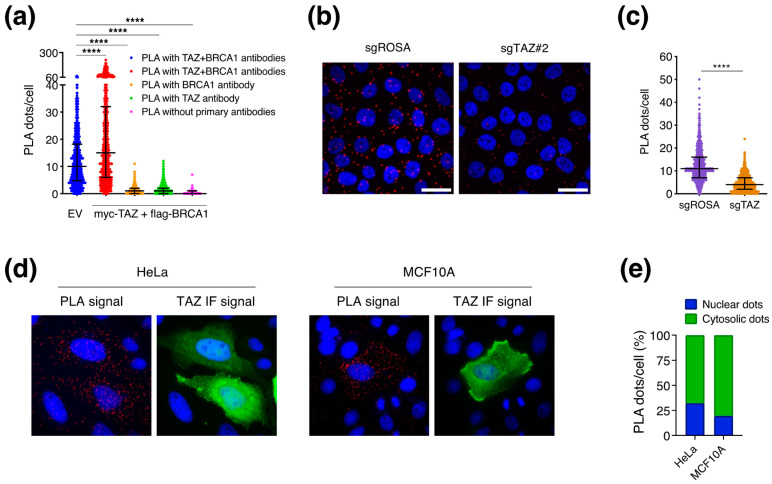
Evidence of the intracellular proximity of TAZ and BRCA1. (**a**) HeLa cells were transfected with the pQCX–TAZwt–Myc–tag and the pMH–BRCA1–FLAG plasmids to overexpress TAZ and BRCA1. After 48 h from the transfection, sub-confluent cells were fixed and processed. Dot plot showing the quantification of the PLA–foci per cell. Cells were stained with only one primary antibody (TAZ or BRCA1) or with secondary probes only (no primary antibodies) as technical negative controls. Raw data were analyzed by ImageJ software. Cell area was measured by an ImageJ plugin that predicts polygonal shapes around the DAPI signal. Sample size: 590 cells per condition. Mann–Whitney test was applied to run statistical analysis: **** *p* value <0.0001. (**b**,**c**) MCF10A–rtTA–Cas9–sgROSA and –sgTAZ (#2) cells were seeded in six wells on glass coverslip in presence of 1 µg/mL of doxycycline to induce Cas9 expression. After 72 h sub-confluent cells were fixed and processed. (**b**) Representative micrographs at 100× magnification of PLA–foci detected in MCF10A cells wild type for TAZ (sgROSA) or knock-out for TAZ (sgTAZ). In blue the nuclei stained with DAPI, in red the PLA signal showing TAZ–BRCA1 proximity. (**c**) Dot plot of the number of PLA foci per cell. Sample size: 635 cells per condition. Mann-Whitney test was applied to run statistical analysis: **** *p* value < 0.0001. (**d**,**e**) HeLa and MCF10A cells were transfected with plasmids to overexpress TAZ and BRCA1. After 48 h, sub-confluent cells were fixed in PFA 4% and processed for IF and PLA analysis. (**d**) Representative images at 100× magnification: in blue Dapi stained nuclei, in red PLA-dots, in green TAZ overexpression detected by IF. (**e**) Bar plot showing the fraction of PLA dots per cell co-localizing with Dapi signal (nuclear proximity) or with TAZ signal but not with the Dapi (cytosolic proximity). Images were analysed with the ImageJ software. Sample size: 94 HeLa cells and 40 MCF10A cells overexpressing TAZ.

**Figure 5 cells-12-02431-f005:**
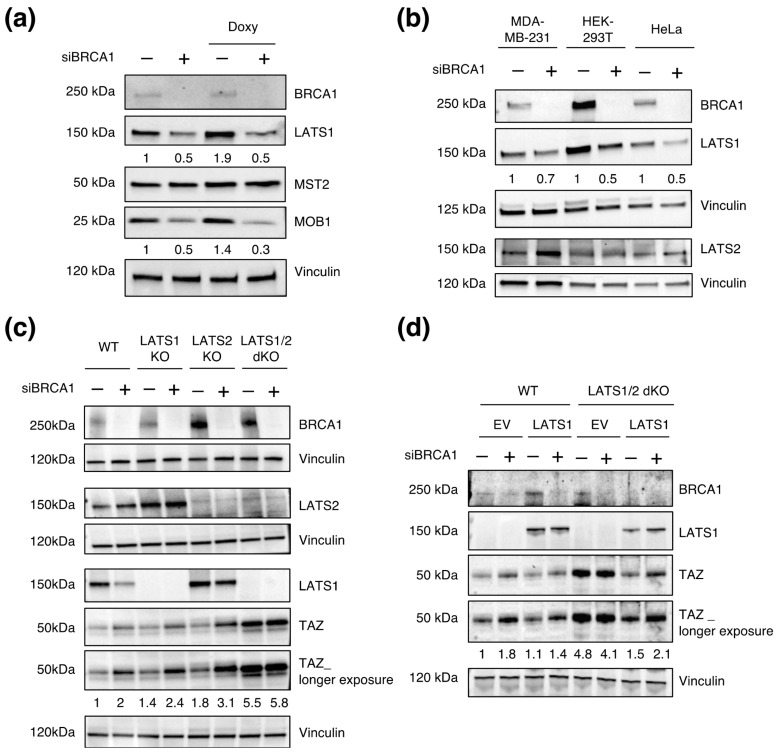
Hippo signaling is epistatic over BRCA1 regulation. (**a**,**b**) WB analysis of Hippo pathway components (LATS1, MOB1, MST2, and LATS2) in (**a**) MCF10A–pSLIK–TAZS89A, and (**b**) MDA–MB–231, HEK–293T, and HeLa. Cells were transfected with siBRCA1 (#458) or with a non–targeting siRNA (siC) as control. After 48 h, cells were lysed for protein extraction. Vinculin was used as loading control. (**c**) WB analysis of HEK–293A cells wild-type (WT) or KO for LATS1, LATS2 or both (dKO). Cells were transfected with siBRCA1 (#458) or with a non-targeting siRNA (siC) as control for 48 h. Vinculin was used as loading control. (**d**) WB analysis of HEK–293A cells wild type (WT) or double knock-out for LATS1,2 (dKO). Cells were first transfected siRNA against BRCA1 (#458) or with a non-targeting siRNA (siC), and then, 24 h later, with pEGFP C3–LATS1 plasmid or an empty vector (pEGFP–EV). Cells were collected 48 h from the first transfection and processed for protein analysis. Vinculin was used as loading control. The normalized densitometric values of the WB bands are reported below the WB snapshots.

**Figure 6 cells-12-02431-f006:**
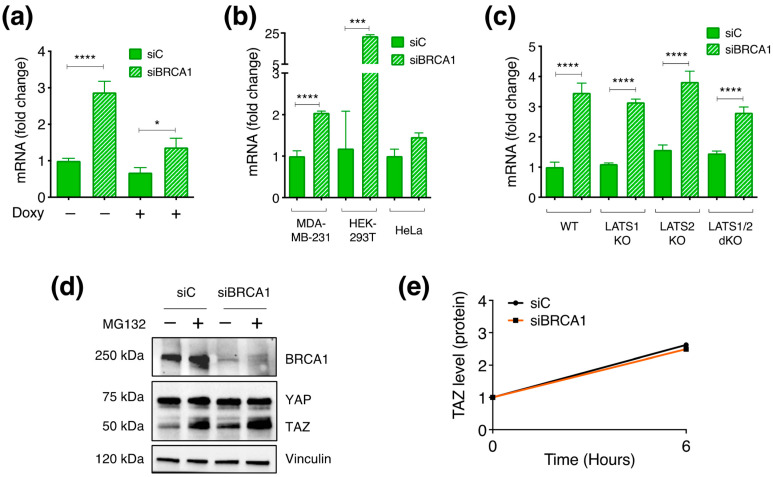
Regulation of TAZ by BRCA1 is mainly by post–translational. (**a**–**c**) RT–qPCR analysis of (**a**) MCF10A–pSLIK–TAZS89A–8xTEAD–LUC cells, (**b**) MDA–MB–231, HEK–293T and HeLa cells, and (**c**) HEK-293A cells wild type (WT), single KO for LATS1 and LATS2 or dKO. cDNA levels were normalized to either GAPDH or RPLPO housekeeping genes and expressed as fold change. *T*–test was applied to evaluate the statistical significance: * *p* value < 0.05 *** *p* value < 0.005 **** *p* value < 0.001. (**d**,**e**) WB analysis of confluent MCF10A cells treated with 5µM of the proteasome inhibitor MG132 for 6 h. (**e**) Densitometric analysis of the WB shown in (**d**). TAZ signal was normalized to Vinculin signal and expressed as fold change compared to the not-treated condition (-).

**Figure 7 cells-12-02431-f007:**
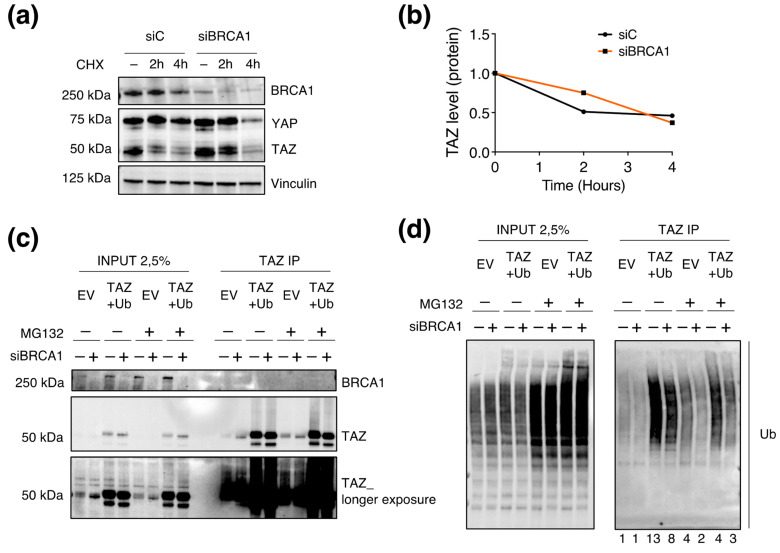
BRCA1 regulates TAZ ubiquitylation (**a**,**b**) WB analysis of confluent MCF10A transfected with siBRCA1 (#458) or with a non-targeting sequence (siC), and treated with CHX for the indicated times. (**b**) Densitometric analysis of the WB shown in (**a**). (**c**,**d**) Analysis of TAZ ubiquitylation by immunoprecipitation (IP) and WB analysis. HEK–293A cells were transfected with siBRCA1 (#458) or a mock siRNA (siC). After 48 h, cells were transfected with plasmids encoding for TAZ (pMSCV-HA-TAZ) and Ubiquitin (pcDNA3–FLAG–Ub). At 72 h from the first transfection, cells were treated with 5 µM of proteasome inhibitor MG132 for 6 h. 2.5% of the immunoprecipitated lysate (Input) was loaded to assess the fold enrichment in TAZ IP. The normalized densitometric values of the WB bands are reported below the WB snapshots.

**Figure 8 cells-12-02431-f008:**
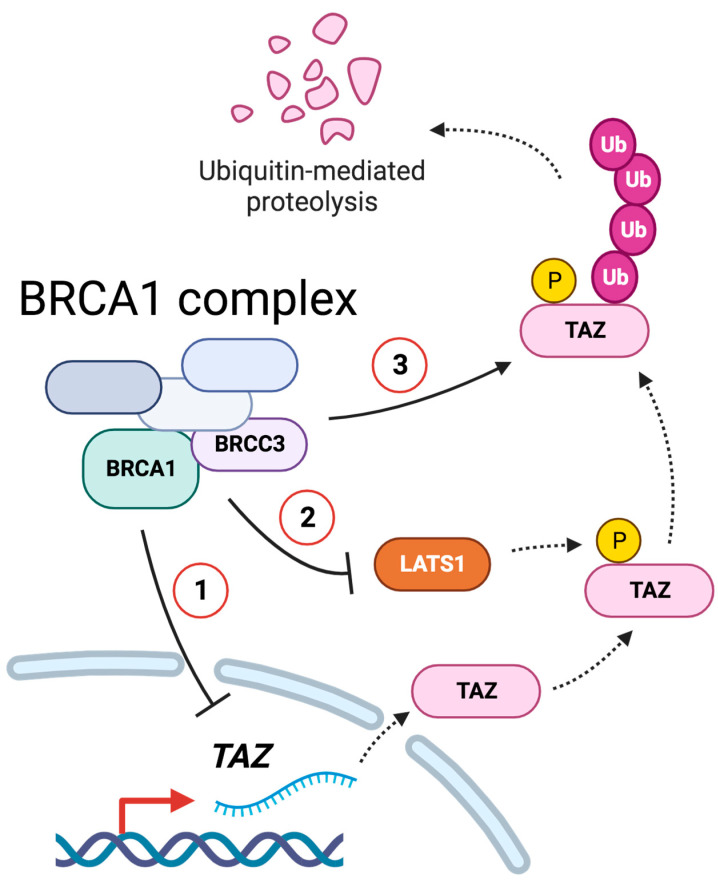
Model of how BRCA1 and BRCC3 control TAZ. BRCA1/BRCC3 control TAZ activity by modulating (1) TAZ transcription (2) LATS1 protein level (3) TAZ ubiquitylation and proteasomal degradation. Created with BioRender.com (accessed on 4 October 2023).

## Data Availability

Data are available upon request.

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
