# Peer review of "Identification of BRCC3 and BRCA1 as Regulators of TAZ Stability and Activity"

_cells, 2023, doi:10.3390/cells12202431_

Round 1

Reviewer 1 Report

The manuscript " Title: Identification of BRCC3 and BRCA1 as regulators of TAZ 2 stability and activity" by Sberna et al. reports that BRCA1 regulates TAZ post-transcriptionally by ubiquitination, not TAZ synthesis nor localization, in the cell cytoplasm. The authors have also reported that BCRA1 is involved in the Hippo pathway to modulate TAZ. Overall, this is a methodical, well-organized manuscript that should interest readers of cancers. However, a few issues should be addressed before the manuscript is accepted for publication. BRCC3 is not as thoroughly analyzed as BRCA1 in this paper. Please include BRCC3 analysis in the discussion part to emphasize the point the title presents. Remove lines 638-641.

Reviewer 2 Report

The manuscript did a genetic screening to identify regulators of TAZ transcriptional activity in mammary epithelial cells and described an initial study of the potential regulation of TAZ by BRCA1 and other components of the BRCA complex.  In general, this study is well-designed and well-written but the methodology description needs to be improved

1.     Suggest adding the statistical method part in section 2 and explain how and why the T test and Mann-Whitney tests were chosen and applied?

2.     Can you explain why there is no T test result for CTGF for HEK-293T in Figure 2(i)? Is it because it is not statistically significant but those two groups look different visually?

Reviewer 3 Report

 Dear editor,

In the manuscript entitled “Title: Identification of BRCC3 and BRCA1 as regulators of TAZ stability and activity” the authors used several approaches to support the hypothesis that TAZ expression is regulated by BRCA1.

Even though the study is well conducted, and the manuscript well written in most part, some minor issues raised the following concerns:

1- In the first paragraph of the Discussion section is written “Authors should discuss the results and how they can be interpreted from the perspective of previous studies and of the working hypotheses. The findings and their implications should be discussed in the broadest context possible. Future research directions may also be highlighted.” This is probably an excerpt from a previous reviewer's comment from elsewhere. Please remove it.

2-The authors must include a final figure that summarizes the hypothetical pathway of TAZ regulation based on the findings presented in this manuscript.

3-In fact, the suggestion of the previous reviewer make sense, since the authors really should discuss how the results can be interpreted from the perspective of previous studies. The second paragraph of the discussion lacks references, and it is in most part, a summary of the main results.

 If the authors consider improving the discussion, maybe the manuscript fits the quality standard of the journal.

Reviewer 4 Report

Sberna et al. have presented a study revealing that BRCA1 and other components of the BRCA1 complex serve as negative regulators of TAZ, a transcriptional co-activator associated with the Hippo signaling pathway. Through a combination of genetic screening and an array of cell-based experiments, the authors identified and validated several BRCA1 complex components as regulators of TAZ. Furthermore, they proposed that BRCA1 primarily suppresses TAZ levels and activity through post-translational mechanisms. To substantiate their hypothesis, the authors conducted a series of well-designed experiments. Their findings are not only intriguing but also hold the potential to captivate researchers in the realms of TAZ, DNA damage/repair, and cancer biology fields. However, there are some areas in the manuscript that require improvement to meet the standards for publication in Cells.

Major points

-         The authors utilized TAZS89A as a reporter throughout the manuscript. They should address if there is any (gain of function) effects associated with TAZS89A overexpression.  

-         In Fig 1, the authors should specify the exact figure numbers for the results  (e.g. In line 348, Figure 1  à  Figure 1d ;  Line 352 Figure 1 à Fig 1e;  Line 359 Fig 1d

-         In Fig 1d and the accompanying text, revise the number of up and down-regulated genes. The number in the text is inconsistent with the number in the figure (92 UP in Fig, 91 UP in text. 45 down/lethal in figure, 43 in text).

-         Quantification of (key) WB bands throughout the manuscript should be added for better comprehension.

-         The quality of some WB images needs improvement, especially BRCA1 (e.g. Fig 5d). Also, TAZ bands patterns are not consistent (e.g. Fig 5c weak lower band vs 5d weak higher band)

-         In Fig 2b, the authors mentioned ‘there was a consistent increase of TAZ protein, observed for both the endogenous TAZ and the ectopically expressed TAZS89A’. They need to label the TAZS89A and endogenous TAZ protein on the blot, as there are two bands.

-         In Fig3, the localizations of TAZ by two antibodies are quite different. Is it normal? Do they detect different isoforms?

-         In Fig 4 PLA assay, specify which TAZ antobody was used for the PLA. It was not specified neither in the materials/methods nor in supplementary table. This is important as C22 (more cytoplasmic) and C188 (nucleus) staining patterns differ.

-         Fig 6 is somewhat confusing. In Fig 6a, the authors should clarify if they measured TAZ level by qPCR (add it in title or y-axis). Also need to clarify why lower TAZ level in the Doxy + than Doxy -.

-         Add a model figure or cartoon (pathway) as a main figure. This will help readers better understanding of their findings

Minor points

-         In Fig 1, they should address how/what criteria they determined the thresholds for viability and reporter assay. Additionally if any statistical analysis was performed, they should provide details.

-         In Fig 2, they should show the KD efficiency (quantification) of BBRC2 in supp figure (similar to Fig 2c). judging from the WB, the KD efficiency may not be strong

Round 2

Reviewer 3 Report

The reviewer´s queries were properly answered. Thus, this reviewer RECOMMENDS this manuscript for publication in Cells.